# Synthesis of Covalent Organic Frameworks (COFs)-Nanocellulose Composite and Its Thermal Degradation Studied by TGA/FTIR

**DOI:** 10.3390/polym14153158

**Published:** 2022-08-02

**Authors:** Chunxia Zhu, Shuyu Pang, Zhaoxia Chen, Lehua Bi, Shuangfei Wang, Chen Liang, Chengrong Qin

**Affiliations:** 1Guangxi Key Laboratory of Clean Pulp & Papermaking and Pollution Control, School of Light Industrial and Food Engineering, Guangxi University, Nanning 530004, China; 2016301052@st.gxu.edu.cn (C.Z.); 2116301042@st.gxu.edu.cn (S.P.); 2116391003@st.gxu.edu.cn (Z.C.); wangsf@gxu.edu.cn (S.W.); qin_chengrong@163.com (C.Q.); 2Xingjian College of Science and Liberal Arts, Guangxi University, Nanning 530004, China; bilele@163.com

**Keywords:** covalent organic frameworks, aldehyde cellulose nanocrystals, mechanochemistry, TGA/FTIR

## Abstract

At present, the synthesis methods of crystalline porous materials often involve powder products, which not only affects the practical application but also has complex synthesis operations and limited scale. Based on the mechanochemical method, we choose COF-TpPa-1, preparing TpPa-1-DANC composites. Covalent organic frameworks (COFs) are a kind of crystalline material formed by covalent bonds of light elements. COFs possess well pore structure and high thermal stability. However, the state of synthesized powders limits their application. Cellulose nanocrystals (CNCs) are promising renewable micron materials with abundant hydroxyl groups on their surface. It is possible to prepare high-strength materials such as film, water, and aerogel. Firstly, the nanocellulose was oxidized by the sodium periodate method to obtain aldehyde cellulose nanocrystals (DANC). TpPa-1-DANC not only had the crystal characteristic peak of COFs at 2θ ≈ 5° but also had a BET surface area of 247 m^2^/g. The chemical bonds between COFs and DANC formed by Schiff base reaction appeared in FTIR and XPS. The pyrolysis behavior of the composite was characterized by TG-IR, which showed that the composite had good thermal stability. With the advantages of nanocellulose as a material in every dimension, we believe that this method can be conducive to the large-scale synthesis of COFs composites, and has the possibility of multi-form synthesis of COFs.

## 1. Introduction

Porous organic materials have received much attention due to their excellent performance in absorption, catalysis, biomedicine, energy storage, and other valuable applications [1,2,3]. Covalent Organic Frameworks (COFs) are crystalline polymers fabricated by covalent bonds of light elements (C, N, B, H, etc.) [4]. COFs were first synthesized in the form of boron oxygen bonds by Yaghi [5] using a hydrothermal method in 2005, opening a new door to porous materials. COFs materials possess advantages including robust and ordered porous structure, modifiability, low density, and high thermal stability [6]. It has become a research hotspot in the fields of gas adsorption and separation [7], catalysis [8], and energy storage [9]. Nevertheless, due to the crystallization characteristics and rigid structure of COFs, most of the as-synthesized COFs are insoluble powders. Firstly, COFs materials are difficult to process and shape [10]. Secondly, powder makes it difficult for COFs to be recycled after use, which limits the application of COFs materials. Thirdly, it is not conducive to the development of COFs composites. In this current situation, it would be highly desirable to form a new method.

Cellulose is an ample natural polymeric material on the earth and is derived from plants, marine creatures, and bacteria [11,12,13], etc. Cellulose nanofibers (CNFs) possess advantages including tensile strength and easy membrane-forming [14] that can be used as composite skeleton or substrate. With outstanding biocompatibility [15,16], nanocellulose currently has achieved a variety of nano-adopting forms, including carbon nanotubes [17] and metal organic frameworks (MOFs) [18], etc. Qian et al. [19] found that adding CNF can significantly improve the mechanical properties of MOFs/cellulose composites. Wan et al. [20] synthesized Pd@COF/NFC composite membranes in situ on modified nanocellulose membranes based on a hydrothermal method. However, considering the compatibility of the two-dimensional (2D) layered structure of COFs with substrates, it remains a challenge to fabricate defect-free COFs layers on substrates by in situ hydrothermal methods [21]. Therefore, Abdul et al. [22] developed a strategy of in-situ solid phase doping of carbon nanofibers (CNFs) to prepare COF-CNF hybrid films by mechanochemical methods. The common preparation methods of COFs include the hydrothermal method [23,24] and the microwave-assisted method [25,26], which usually have complicated operation, harsh conditions, and long reaction times and accumulate the organic solvent waste. It is necessary to explore suitable solvent systems and catalysts and conditions such as the ratio of monomers [27]. From the perspective of ecology and energy consumption, it is exigent to explore an approachable and workable strategy of COFs with simpler and faster operation.

Comparatively, the mechanochemical synthesis of COFs is an energy-efficient and low-cost synthesis method [28]. The mechanochemical method can be run under a shorter cycle and milder condition of no or less solvent, which has the feasibility of environmental friendliness and the possibility of expanding the scale of material production [29]. In mechanochemical reactions, such as ball milling, the size of solid particles decreases, and the accumulated potential energy leads to the chemical reaction of substances [30]. The mechanochemical method is currently used in the synthesis of nanomaterials or compounds, such as nanoparticles, zeolites, porous carbons, metal complexes, metal organic frameworks (MOFs), etc. [31,32]. Khayumc et al. [33] used synthetic flexible COF flakes as electrode materials, and the use of mechanochemistry could allow one to avoid the use of binders or additives. Due to the outstanding capacity of COFs, considering the plasticity and scalability of mechanochemical methods and the multidimensional nature of nanocellulose, a simple solvent-free method is explored to synthesize covalent organic framework-nanocellulose composites.

To explore efficient, environmentally friendly, and easy-operate synthetic methods, mechanochemical methods were selected to prepare covalent organic framework (COF-TpPa-1) and covalent organic framework-nanocellulose composite (COF-DANC). Aldehyde group modified nanocellulose crystals were selected to synthesize composites with a certain degree of crystallinity and specific surface area, and the preparation and characterization of covalent organic framework-nanocellulose composites were explored. At the same time, the thermogravimetric analysis combined with infrared spectroscopy was used to analyze the pyrolysis products and yield analysis, and to explore the feasibility of mechanochemical synthesis of composite materials.

## 2. Materials and Methods

### 2.1. Materials

P-phenylenediamine (Pa-1) was purchased from Sigma, 98%; Trialdehyde phloroglucinol (Tp) was purchased from McLean, Shanghai, China, 97%; Acetone was purchased from Chengdu Kelong Chemical Co., Ltd., Chengdu, China, AR. Microcrystalline cellulose (MCC) was purchased from Sinopharm Chemical Reagent Co., Ltd. Shanghai, China, column chromatography.

P-toluenesulfonic acid (PTSA) was purchased from Tianjin Damao Chemical Reagent Factory, Tianjin, China, AR; N,N-dimethylformamide (DMF) was purchased from Tianjin Zhiyuan Chemical Reagent Co., Ltd., Tianjin, China, AR; Sulfuric acid (H_2_SO_4_) was purchased from Ningbo Xinzhi Biotechnology, Ningbo, China, analytically pure; sodium periodate (NaIO_4_) was purchased from Aladdin reagent, Shanghai, China, AR; Ethylene glycol was purchased from Thermo Fisher Scientific, Waltham, MA, USA, AR. All reagents and solvents are commercially available and used without further purification.

Dialysis bag MD77 was purchased from United Carbon, MWCO 3500. The portable grinder was purchased from Shanghai Wanbo Bio, Shanghai, China, Mini-2-5.

### 2.2. Preparation of Dialdehyde Cellulose Nanocrystals (DANC) by Periodic Acid Oxidation

Cellulose nanocrystals (CNCs) were prepared from microcrystalline cellulose (MCC) by sulfuric acid hydrolysis [34]. Oxidation of CNCs suspension was conducted using sodium periodate to obtain 2,3-dialdehyde nanocrystalline cellulose (DANC) [35]. 30 g CNCs suspension (0.5 g absolute dry) and a certain amount of NaIO_4_ (molar ratio 1:4) were added to a 100 mL flask. The flask was covered with aluminum foil to avoid light to avoid the decomposition of NaIO_4_ in light. The oxidation reaction was carried out by stirring at 70 °C for 180 min. Then, 10 mL of ethylene glycol was added and allowed to react for another 30 min to remove unreacted NaIO_4_ to terminate the oxidation reaction. The suspension was dialyzed (MWCO 3500) for one week in deionized water, and was stopped when the conductivity of the filtrate was less than 2 μs/cm on a conductivity meter. The yield was calculated, and the suspension was stored at 4 °C for standby.

### 2.3. Preparation of TpPa-1-DANC

Covalent organic framework nanocellulose composite (COF-DANC) was prepared in a portable grinder. Then, 475 mg of P-toluenesulfonic acid (PTSA), 47.5 mg of p-phenylenediamine (Pa-1), and 100 mg of 2,3-dialdehyde nanocrystalline cellulose (DANC) were added into the centrifuge tube, and two 3 cm diameter zirconia balls were ground in a portable grinder for 5 min. To this 0.3 mmol (63 mg) of trialdehyde phloroglucinol (Tp) was added and ground for 10 min, then grinding was continued with 5.5 mmol (100 μL) deionized water for 5 min. The mixture was removed from the centrifuge tube, then heated in the oven at 170 °C for 60 s. After cooling to room temperature, it was filled with water to wash away PTSA, and then ultrasonic washed with DMF and acetone 2–3 times to remove unreacted monomer and oligomer impurities. After methanol soxhlet extraction for 12 h and vacuum drying at 60 °C for one night, the powder was COF-DANC. When the addition amount of DANC was 50 mg, the material was named COF-DANC-A.

### 2.4. Characterization

Fourier transform infrared (FT-IR) spectra were recorded with Nicolet iS50 (Thermo Fisher Scientific) with a Diamond ATR (Golden Gate, MA, USA) accessory in the 600–4000 cm^−1^ region, the sample was vacuum dried at 60 °C overnight before testing. X-ray photoelectron spectrometer (XPS) was tested with ESCALAB 250XI+ (Thermo Fisher Scientific). Solid state NMR (SSNMR) was taken in Agilent 600M (Agilent, Santa Clara, CA, USA), chemical shifts were expressed in parts per million (δ scale). The X-ray Diffraction (XRD) was performed with a SmartLab 3KW (Rigaku, Tokyo, Japan) for Cu K*α*
radiation (λ = 1.5406 Å). The operating voltage and current were 40 kV and 40 mA in the range of 3°–40° with the scanning speed of 2°/min and the step of 0.01°. The samples were dried and gently ground into suitable powder for the XRD test. The crystallinity index (C_r1_) of nanocellulose samples can be calculated by Equation (1). [36,37]. In the Segal method, the C_r1_ of nanocellulose can be interpreted by main crystalline peak I_002_ (2θ ≈ 22°) and amorphous peak I_am_ (2θ ≈ 18°). Measurements were repeated at least twice.
(1)$$C_{r1}=\frac{I_{002}{-I}_{\mathrm{am}}}{I_{002}},$$

Scanning electron microscopy (SEM) images were recorded using a Hitachi SU8220 (HITACHI, Tokyo, Japan) with tungsten filament as an electron source operated at 10 kV. Then, 1 mg of sample was dispersed in 10 mL of isopropanol and sonicated for 60 min, then the dispersion was dropped casting multiple times on the silicon bottom specimen stage, and Au sputtering was conducted before analysis. Transmission scanning electron microscopy (TEM) analysis was performed using HITACHI HT7700 at an accelerating voltage of 200 kV. For TEM sample preparation, the sample was dispersed in isopropanol, and the supernatant was dropped on the copper grid (200 mesh). After air-drying at room temperature for several minutes, the sample was negatively colored with phosphotungstic acid dye under dark conditions. After 30 min, the excess dye was absorbed with filter paper.

The specific surface area of the samples was measured by nitrogen adsorption–desorption experiments at 77 K performed on the ASAP 2460 (Micromeritics, Atlanta, GR, USA). The samples were outgassed under vacuum (150 °C, −0.1 MPa) overnight prior to the N_2_ adsorption studies. The surface areas were evaluated using the Brunauer–Emmett–Teller (BET) model. Thermogravimetric infrared spectroscopy (TGA-FTIR) experiments were performed using a TGA55 thermogravimetric analyzer (TA, Everett, MA, USA) combined with a Nicolet iS50 Fourier transform infrared spectrometer (Thermo Fisher). The heating rate of the experiment was 15 °C/min, the flow rate of the nitrogen atmosphere was 40 mL/min, and about 10 mg of the sample was taken. The thermogravimetric overflow gas was detected synchronously in the infrared equipment through the connected insulation pipe.

## 3. Results and Discussion

Here, according to the previous report by Karak [38], COF-TpPa-1 can be prepared more easily and on a larger scale by the mechanochemical method (Figure 1a). Microcrystalline cellulose (MCC) introduced aldehyde groups on the surface by the sodium periodate method to obtain 2,3-dialdehyde nanocrystalline cellulose (Figure 1b). Aldehyde-based nanocellulose and organic monomer were added step by step to prepare the TpPa-1-DANC composite. Figure 1c provides a schematic of the synthesis procedures of COF-TpPa-1 and composite with the mechanochemical method.

Ground P-toluenesulfonic acid (PTSA), with DANC and Pa-1, was gradually added to Tp-1 and a small amount of water, and grinding was continued until evenly mixed and then the mixture was heated. The aldehyde group on DANC reacted with the amine group on Pa-1 to form Schiff base C=N, which helps DANC connect to the COF nanosheet. PTSA can promote Schiff base reaction and contribute to the formation of imino groups in COF.

### 3.1. Morphology Analysis of CNCs and DANC

As shown in Figure 2, the SEM pictures showed that microcrystalline cellulose (MCC) (g) was acidolysis to obtain cellulose nanocrystals (CNCs) (h), and then the 2,3-dialdehyde nanocrystalline cellulose (DANC) was modified by sodium periodate method. As can be seen in the following (i), a wide needle DANC with a width of 20 ± 7 nm (average ± one standard deviation) and a length of about 200 ± 50 nm (average ± one standard deviation), and the dispersion degree was better than NCC. According to previous studies, the surface modification of nanocellulose [39] can reduce the agglomeration phenomenon, so as to improve the dispersion of CNCs and the interface interaction of the substrate [40]. By introducing aldehyde groups to the surface of CNCs, a well-dispersed DANC can be obtained. In Figure 3d the nanocellulose in XRD presented the main peaks at around 14, 16, and 22°, belonging to the typical characteristics of cellulose I-type structure, corresponding to 101, 1ō1, and 002 crystal planes, respectively. The absorption peak near 34° came from the 004 crystal plane of cellulose I-type structure. It proved that the crystalline structure of DANC does not rearrange after modification. According to Equation (1), the crystallinity indexes (C_r1_) of CNCs and DANC were 0.73 and 0.056. This was because, with the oxidation of sodium periodate, glucopyranose opened the ring, resulting in the change from the crystal surface to the internal structure, so the crystallinity index decreased. Referring to the literature [41], based on the oxidation reaction between the aldehyde group and hydroxylamine hydrochloride, aldehyde content of 7.1 mmol/g was obtained by calculation, which is conducive to the synthesis of nanocellulose matrix composites.

### 3.2. Characterization of COF-TpPa-1 by Mechanochemical Method

From Figure 4a,b, in COFs synthesized by the mechanochemical method, it can be seen that the transverse dimension of COF-TpPa-1 was 10–15 μm. Its lamellar shape corresponds to the layer stacking seen in the TEM of Figure 4c, which is consistent with the results in the literature [38].

The structure of the covalent organic framework is an organic framework with a repeated unit structure, so it has certain predictability and regularity. The crystal structure of COFs and their composites were analyzed by X-ray diffraction (XRD), and the structural information such as the regularity of the framework or the stacking degree of layers of the samples could be known. The XRD spectrum of the TpPa-1 sample prepared by the mechanochemical method was shown in Figure 3d. It can be seen that there was an obvious characteristic peak at 4.9°, corresponding to the (100) plane of covalent organic framework~8.7° and 11.9° are attributed to (200) and (210) reflection planes, respectively. The 2θ = 27° angle belonged to the amorphous peak, which is attributed to the (001) plane п-п Stacking, and layered stacking distance was 3.3 Å. These results were consistent with previous reports on pure covalent organic frameworks [42].

Figure 4d showed the infrared (FTIR) diagram of Tp, Pa-1, and COF-TpPa-1. It can be seen that the C–H characteristic peak 2894 cm;^−1^ of Tp and the C=O characteristic peak 1640 cm^−1^ of carbonyl stretching band disappear. The peak of COF-TpPa-1 at 3400–3500 cm^−1^ was the stretching vibration peak of –OH, and there was no N–H characteristic peak (3296 and 3367 cm^−1^), indicating that Pa-1 was consumed after the reaction. The C–N stretching vibration peak appeared at 1256 cm^−1^, and the secondary amine (N–H) bending vibration peak appeared at 1517 cm^−1^. The absorption peak of the benzene ring skeleton appeared at 1454 cm^−1^. A new peak of C=C appeared at 1585 cm^−1^. The C–H characteristic peak of aldehyde –CHO disappeared at 2820 cm^−1^, indicating that the aldehyde group in the molecular structure reacted, and the characteristic peak of ketone C=O appeared at 1604 cm^−1^, which is in good agreement with previous reports [38]. In conclusion, the synthesized COF-TpPa-1 had the structure of aldehyde C=O structure conjugated with C=C, and the structure of the para-substituted benzene ring and imine bond (C–N). This indicates that the condensation reaction and imine bond formation have occurred, that is, β-ketene amine skeleton formation.

Solid nuclear magnetism (NMR) reflects the chemical environment within the nucleus by combining the internal and surrounding structures of the nucleus. The structural network of COF-TpPa-1 was analyzed by ^1^H NMR and ^13^C NMR, which provided a basis for thermogravimetric infrared analysis and a reference for further optimizing the synthesis process. In the ^1^H NMR diagram of Figure 4e, it can be seen that a strong peak of ~5 ppm was assigned to water molecules from COF-TpPa-1. An obvious signal centered at ~8.0 ppm was assigned to the protons of amino groups, indicating that amino groups and adjacent carbonyls may form strong intramolecular hydrogen bonds and mutual shielding generated by aromatic rings [43]. As shown in Figure 4f, it was the solid-state ^13^C NMR spectrum of COF-TpPa-1. The chemical shift at 146 ppm can be considered as carbon (=CNH–) on enamine, and the signal at 181 ppm came from carbonyl carbon (–C=O). And the 157 ppm of TpPa-1-DANC came from the C=N bond produced by the aldehyde amine condensation reaction [44]. The small peak at 191 ppm was attributed to the carbon atom of the aldehyde group at the end of COF-TpPa-1 in the composite, which is consistent with the FTIR results. It was confirmed that in the covalent organic framework, β-ketene amine bond formation occurred.

### 3.3. Characterization of TpPa-1-DANC Composites by Mechanochemical Method

#### 3.3.1. Physical Properties

TpPa-1 and DANC were compounded by the mechanochemical method. The morphology of TpPa-1-DANC was analyzed by SEM and TEM. In Figure 3d,e of TpPa-1-DANC, the average particle size scale increased to 15 μm. It can be seen by transmission electron microscope that the contact between flake stacked COF and rod-needle-shaped DANC was a transitional blur, rather than a clear interface, which indicates that the two are not going through simple physical blending, but covalent grafting [22].

Compared with TpPa-1, the XRD of TpPa-1-DANC not only showed the characteristic peak of COF but also showed the characteristic peak of (110) crystal plane of cellulose at about 22°, indicating that DANC is successfully incorporated into the TpPa-1 network. The peak at 2θ = 27.22° decreased, which may be due to the fact that the COF grown on the (001) crystal plane was parallel to the direction of cellulose [20].

#### 3.3.2. Chemical Properties

TpPa-1-DANC had an obvious characteristic peak at about 1400 cm^−1^, and about 2900 cm^−1^ came from the C–H bond of 2,3-dialdehyde nanocrystalline cellulose (DANC). The stretching vibration of the C=N bond at 1629 cm^−1^ [45] shows the formation of the imine bond, the weakening of the C–N bond of COFs, the disappearance of C=O of DANC, and the weakening of C–N, which proves that they are successfully grafted through reaction [46].

X-ray photoelectron spectroscopy (XPS) characterization technology irradiates the sample surface with X-ray photons. According to the analysis of the electronic energy distribution of the elements on the sample surface, judge whether the relevant chain chemistry is formed, and investigate the content and grafting of COFs composite. XPS analysis of TpPa-1 and its composites are shown in Figure 3. It can be seen from the full spectrum f) that there is an obvious N peak in TpPa-1. With the addition of DANC, the proportion of the N peak decreases slightly. Measuring the content ratio of the C 1s atom to the N 1s atom of TpPa-1 (C/N = 8.66), it was lower than that of COF-DANC-A (C/N = 9.54). And this was consistent with the experimental data of materials by EDS in Table 1. This proved the existence of 2,3-dialdehyde nanocrystalline cellulose in the obtained material. Under the conditions of the pure grinding method, the N content was low, which may be due to the agglomeration of COF-TpPa-1 powder and DANC powder without catalyst, and the N peak was not characterized when the material surface is scanned by XPS.

Further high-resolution C spectrum analysis of the sample showed that 284.50, 286.08, and 288.74 eV belong to (sp_2_) C=C, C–O/C–H, and C=O, respectively in the C 1s high resolution of TpPa-1 [47]. N 1s appeared at 399.47 eV and 400.06 eV, and the peaks belonging to Ar–NH_2_ and Ar–NH–C were divided. After loading, TpPa-1-DANC had a new peak of 400.09 eV C=N [48]. It was proved that the reaction of TpPa-1 and DANC produces a C=N bond through covalent grafting.

#### 3.3.3. Thermal Analysis

The thermogravimetric-infrared combined technology connects the thermogravimetric analyzer and infrared analysis through the insulation pipe and analyzes the combined spectrum, including a thermogravimetric diagram, infrared diagram of each time (temperature), thermogravimetric infrared three-dimensional spectrum, and intensity change diagram with time. Thermogravimetric infrared technology obtains the thermal stability and purity of the material by synchronously analyzing the infrared spectrum of the polymer at the weight loss temperature, which is an effective analysis method for the current situation of low synthesis of COFs. In the experiment, the infrared three-dimensional spectrum of pyrolysis products obtained by TGA-FTIR technology at the heating rate of 15°/min shows the corresponding characteristic absorption peaks in the fixed wave number range under the pyrolysis temperature range.

The thermogravimetric curve and thermogravimetric infrared three-dimensional diagrams were shown in Figure 5. The TpPa-1 in Figure 5a had a pyrolysis temperature of 330 and 444 °C with a pyrolysis residue of about 40%. The temperature of the maximum decomposition rate decreased with the increase of DANC, similar to the literature [49]. In Figure 5c, the first pyrolysis peak of TpPa-1-DANC was larger than the area of COF because the addition of DANC increased the total carbon content of the material. The results showed that the composites with high COFs content had advantages in thermal stability. As the proportion of COFs increased, the ash component of composites increased, which is due to the gradual aromatization of COFs residues during carbonization. Therefore, the smaller the proportion of COF in the complex, the residue will decrease with the redduction of aromatics. It has been reported in the literature [18] that the ash content of pure MOF powder TGA is 40.79%, increasing by 20–50% with the loading of MOF in CNC–CMC aerogel powder, thus increasing the thermal stability of composites with the loading of MOF in aerogel. Compared to the thermogravimetric curves of TpPa-1-DANC and DANC, the maximum weightlessness rate of composites was higher than that of DANC, suggesting that the stability of DANC was influenced by the combination of COFs and the residual rate of the composites at high temperature was higher than 20% of DANC. This was because the COFs have a stable aromatic ring, which is more stable than the ring structure of hemiacetal at high temperatures.

The thermogravimetric curves of the composite TpPa-1-DANC still showed two major pyrolysis peaks similar to those of COFs. With the addition of DANC, the initial decomposition temperature of the composites was lower than that of COFs, close to that of nanocellulose, and the maximum decomposition rate remained at about 300 °C, but the decomposition range became increased, thus effectively increasing the thermal stability of the nanocellulose composites. DANC had a significant mass loss at 100 °C due to its strong absorptivity and a significant breakdown of the substance at 200 °C. At 11.6 min, it was about 150 °C, at which point the product has partial water and organic solvents, such as DMF, with characteristic peaks around 2900–3000, 1700, and 1000 cm^−^ corresponding to C–H stretching vibrations, carbonyl C=O double bond telescopic vibrations, and C–O and C–C skeletal vibrations within the C–H in-plane bending vibration of DANC [49].

As shown in Figure 5c, this meant that TpPa-1-DANC has higher thermal stability than DANC. In the temperature range of 300–450 °C, glycosidic bonds, partial C–O, and C–C bonds are broken. The maximum weight loss rate and 600 °C residue rate of TpPa-1-DANC were higher than those of DANC. This was because COFs have stable aromaticity at high temperatures, which is more stable than the ring structure of semi acetals and plays a positive role in the wide application of COFs materials.

According to Beer’s law, the intensity of characteristic absorption peak larger, that is, the higher the absorbance, and the higher relative content of gas components in the total gas. Figure 6 showed the thermogravimetric curve of COF-TpPa-1 at a heating rate of 15 °C/min and the FTIR spectrum of pyrolysis products at different temperatures. It can be seen from the infrared image that pyrolysis is divided into three steps:At about 100 °C, it can be seen in Figure 6 that at 100 (5 min) and 137 °C (7 min), the characteristic peak of water increased at 3400–3700 and 1500–1700 cm^−1^, which may come from the water on the material surface. In addition, there was a certain time error in the delay of gas transmission.At about 19 min (and 300 °C), the absorbance of CO increased slightly. The appearance at 1380 cm^−1^ indicated the vibration of the C–C skeleton. By 30.6 min, a large amount of CO_2_ was produced.The third stage was about 400 °C, which shows the large-scale cracking of the frame. With the increase in temperature, the peak value of 2300 cm^−^ increased. The maximum weight loss peak was about 40.2 min. The absorption peak at 3500–3700 cm^−^ showed that the gas still contains water. This means that when the temperature reached 400 °C, the composite began to decompose.

The absorbance of the corresponding CO_2_ reached the maximum, indicating that the material has good thermal stability at about 300 °C.

Generally speaking, pyrolysis at low temperatures comes from untreated solvents or oligomers in COFs. This is consistent with the excellent thermal stability of COFs, so there is no detection of branch chain fracture or frame cracking of COFs at low temperatures. Therefore, the thermogravimetric infrared images of COFs after methanol extraction are compared to judge whether the COFs material is clean and whether the pores are clean. In contrast, the extraction of COF-TpPa-1 in advance showed similar thermogravimetric performance in three cracking stages.

The difference was that the characteristic peak of CH_4_ appears in the decomposition process. As shown in Figure 7, alkanes begin to be produced at 4.86 min, and the CH_4_ characteristic peak appears at 2970–2900 cm^−^. With the increase in temperature, the absorption peak of methane increased, reaches the absorption peak at 7.3 min, and disappeared at about 19 min. The change in the CH_4_ peak was similar to that of 1390 and 1032 cm^−^ characteristic peaks. The gas produced in this temperature range was considered to be coming from DMF (boiling point is 157 °C), and characteristic peaks appeared around 2900–3000, 1390, 1700, and 1000 cm^−^, corresponding to C–H stretching vibration, carbonyl C=O double bond stretching vibration, C–H plane bending vibration and C–O and C–C skeleton vibration of DMF [49].

XRD and BET analysis showed that the TpPa-1-DANC composite maintained good crystallinity and specific surface area of TpPa-1. Considering various harsh environmental requirements in practical application, a thermal stability test is very important. The thermal stability of COFs was obtained by thermogravimetric analysis, which basically had good thermal stability at 300–400 °C.

#### 3.3.4. Structure Analysis of Composites

The structural stability and permanent porosity of the materials were studied by a nitrogen adsorption-desorption experiment at 77 K. As shown in Figure 8a, COF-TpPa-1 showed a steep absorption curve under the low-temperature line (P/P_0_ < 0.5), showing the characteristics of microporous materials. The specific surface area decreases with the increase of the amount of DANC. The BET specific surface area of TpPa-1 prepared by the mechanical grinding method is 359 m^2^/g, which is bigger than 247 m^2^/g when DANC was added. However, they were less than the specific surface area of COFs prepared by the hydrothermal method [50]. It was speculated that the long-range order may be limited due to certain stripping of COFs in the mechanical method. Also the residue of oligomers in the formation process may be caused by insufficient pores for nitrogen adsorption [51]. Pore size distribution analysis of COF-TpPa-1 from the adsorption isotherms indicated that the samples contain micropores with diameters of 1.1 nm (Figure 8b). 

## 4. Conclusions

COF-TpPa-1 and TpPa-1-DANC were prepared by the mechanochemical method from bottom to top. The successful synthesis of the material was proved by various characterizations. The obtained TpPa-1-DANC composite had a certain specific surface area (BET, 247 m^2^/g). The pyrolysis behavior of the material was analyzed combined with thermogravimetric infrared technology. The results showed that TpPa-1-DANC not only improves the specific surface area but also improves the thermal stability. Combined with the film-forming and gel properties of nanocellulose, our experiment provides a basis for the preparation of morphological covalent organic framework composites based on nano cellulose and improves the application feasibility of COFs materials.

## Figures and Tables

**Figure 1 polymers-14-03158-f001:**
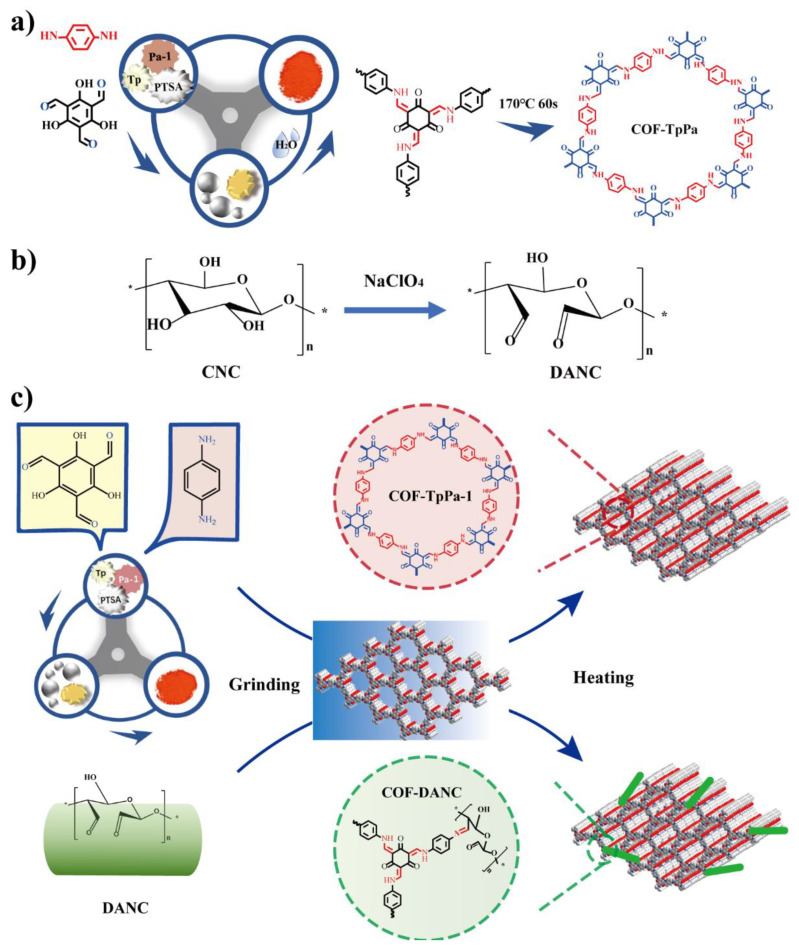
(**a**) Preparation of covalent organic framework COF-TpPa-1 by grinding method; (**b**) Preparation of 2,3-dialdehyde nanocrystalline cellulose (DANC) by sodium periodate method; (**c**) Preparation process of TpPa-1-DANC composite.

**Figure 2 polymers-14-03158-f002:**
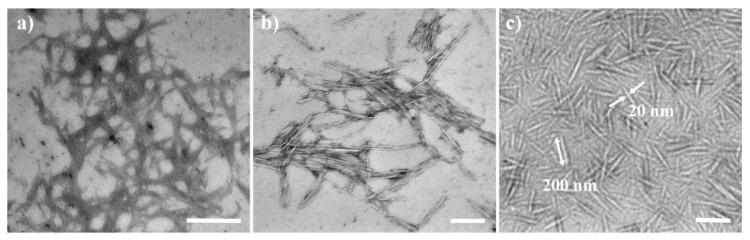
Transmission electron microscope (TEM) of (**a**) MCC, (**b**) NCC and (**c**) DANC; scales of (**a**) is 1 μm, (**b**,**c**) are 200 nm.

**Figure 3 polymers-14-03158-f003:**
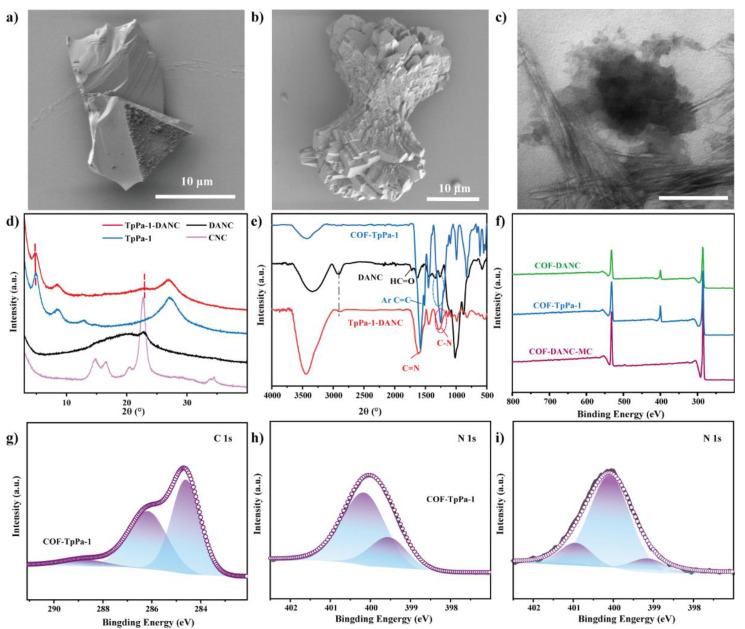
SEM (10 μm) (**a**,**b**) and TEM images (200 nm) (**c**) of TpPa-1-DANC; (**d**) XRD and (**e**) FTIR of TpPa-1, DANC, CNC and TpPa-1-DANC; (**f**) XPS full spectrum of TpPa-1 and TpPa-1-DANC; (**g**) C 1s image of COF-TpPa-1; (**h**) N 1s maps of TpPa-1 and (**i**) COF-DANC.

**Figure 4 polymers-14-03158-f004:**
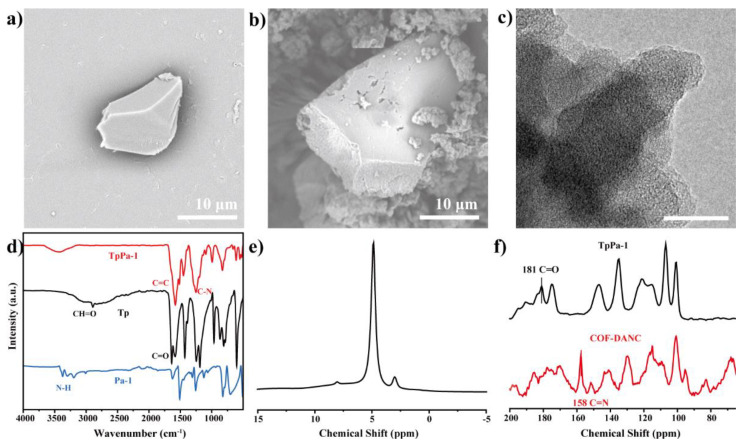
Scanning electron microscope (**a**,**b**) and TEM image (**c**) of TpPa-1; FTIR image (**d**) of TpPa-1 and its monomer; ^1^H NMR (**e**) of COF-TpPa-1; ^13^C NMR (**f**) of COF-TpPa-1 and TpPa-1-DANC.

**Figure 5 polymers-14-03158-f005:**
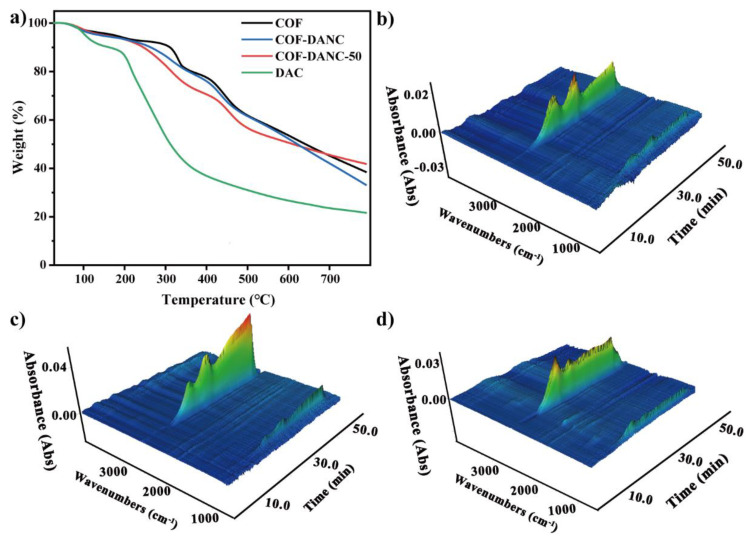
(**a**) the thermogravimetric curve of different materials; (**b**–**d**) the thermogravimetric infrared three-dimensional diagrams of (**b**) COF-TpPa-1, (**c**) TpPa-DANC-50 composites and (**d**) COF monomer mixing.

**Figure 6 polymers-14-03158-f006:**
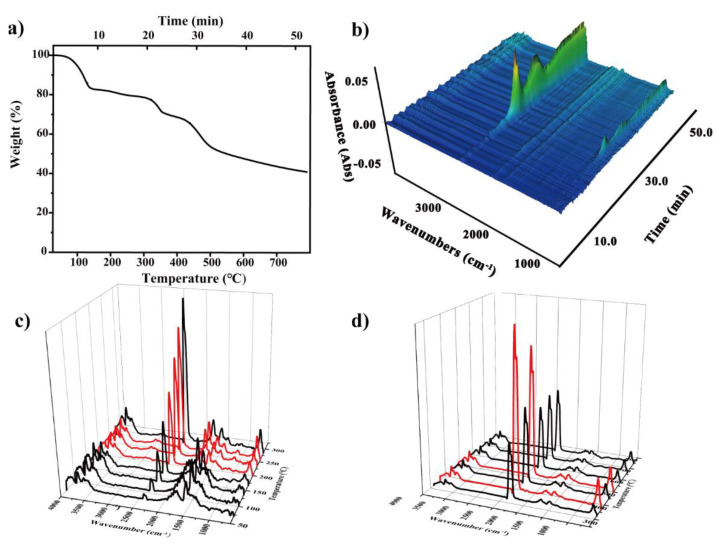
(**a**) Thermogravimetric diagram and (**b**) the thermogravimetric infrared three-dimensional diagram of COF-TpPa-1; Infrared images in the temperature ranges (**c**) 0–300 °C and (**d**) 300–800 °C of COF-TpPa-1.

**Figure 7 polymers-14-03158-f007:**
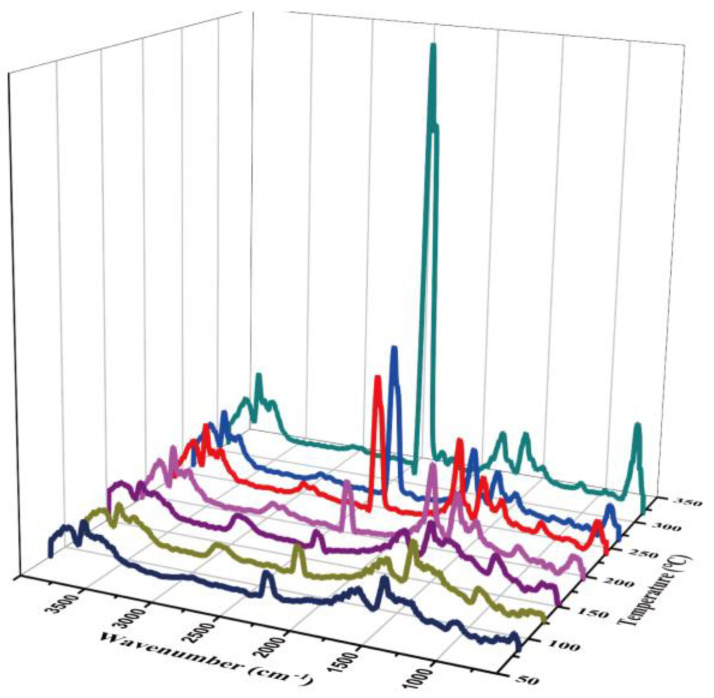
Thermogravimetric infrared three-dimensional diagram of COF-TpPa-1 without Soxhlet extraction in the temperature range of 0–350 °C.

**Figure 8 polymers-14-03158-f008:**
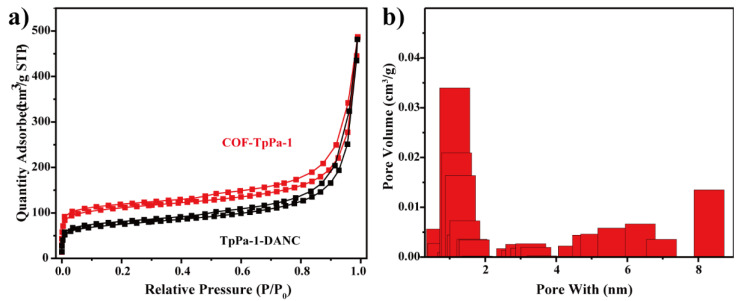
(**a**) Nitrogen desorption curves of TpPa-1 and TpPa-1-DANC; (**b**) Pore size distributions of the pure COF-TpPa-1.

**Table 1 polymers-14-03158-t001:** Summary of XPS atomic mass ratio of different materials.

Element	Atomic %	
	COF	COF-DANC
C 1s	73.15	72.18
O 1s	18.40	20.25
N 1s	8.45	7.57

## Data Availability

The data presented in this study are available in the manuscript’s figure.

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
