# Peer review of "Synthesis of Covalent Organic Frameworks (COFs)-Nanocellulose Composite and Its Thermal Degradation Studied by TGA/FTIR"

_polymers, 2022, doi:10.3390/polym14153158_

Round 1

Reviewer 1 Report

The manuscript entitled “Synthesis of Covalent Organic Frameworks (COFs)-nanocellulose composite and its thermal degradation studied by TGA/FTIR” can be an excellent addition to the cellulose research community and the reader should be benefitted also. However, I have some suggestions as follows-

1.       Comparison figures such as for XRD, and FTIR should normalize for each analysis. If the authors did not do that, they must update their figures.

2.       Authors should include pristine CNC XRD image in figure 4-d. The crystallinity index should change with the modification of CNC to DANC. If not, the authors should include details.

3.       BET analysis was included in the manuscript but, no information regarding pore size and pore size distribution. I am highly recommending including the pore size distribution in this manuscript as this work is dedicated to the organic framework.  

Reviewer 2 Report

I recommend that this paper should be accepted at the present form.

Author Response

We would like to thank the reviewer for your thoughtful review of out manuscript. We have made the effort to improve the clarity and add more relevant references in the revised manuscript. 

Reviewer 3 Report

The topic of the research work and manuscript is really interesting and provides new information. However there are several issues to be addressed towards its quality improvement before thinking of publication.

The manuscript In the abstract, there are several acronyms used which are not explained. In line 36, you should correct “is” to “are”. In line 45, the parenthesis should be placed after “et al.”. The references included and used in this manuscript are quite limited and you could enrich the text in order to describe in detail the state-of-the-art. It is not possible to avoid commenting that the literature used deals with studies of China or Asia. In several points/references there is not a DOI number/URL. The introduction is too short and there are not some (1-3) sentences referring to what is the motivation for the implementation of the specific experimental work. One relevant sentence could be added in the beginning of the abstract following the same pattern/approach (in order the topic to be gradually approached by the reader). Even though you refer to “cellulose nanocrystals”, there is not even one reference on the raw material (wood or forest/agricultural/marine biomass) where the cellulose is coming from. Please add a reference on the potential raw materials and use the recent and relevant manuscript of https://doi.org/10.3390/su132212810 to support such a statement. Did you apply the methodology of some standards concerning the implementation of the materials-text and testing processes? Please provide details on all the equipment cases (XRD for example). You should provide the standard deviation values in the tables, except for the mean values (where possible).

Round 2

Reviewer 3 Report

As I have checked the authors have implemented the proposed changes in the revised version of manuscript towards the improvement of their work. Almost all the changes have been implemented and in my opinion, the manuscript is well-prepared and organized enough to be accepted for publication in this journal. I remain at your disposal for any clarification.